# Volume Deformation of Steam-Cured Concrete with Slag during and after Steam Curing

**DOI:** 10.3390/ma14071647

**Published:** 2021-03-27

**Authors:** Xiaofeng Han, Hua Fu, Gege Li, Li Tian, Chonggen Pan, Chunlei Chen, Penggang Wang

**Affiliations:** 1School of Civil Engineering, Qingdao University of Technology, No.11 of Fushun Road, Qingdao 266033, China; h1449001076@163.com (X.H.); fs215379@163.com (H.F.); ligegel@126.com (G.L.); tlsxf@163.com (L.T.); 2School of Civil Engineering & Architecture, Ningbo Technology University, No. 1 of Qianhu South Road, Ningbo 315100, China; panchonggen@nit.zju.edu.cn; 3Zhejiang Provincial Erjian Construction Group LTD, No. 256 of Chengguan Station Road, Ningbo 315100, China; cjj4629@sina.com; 4Research Centre of Concrete Technology in Marine Environment of Education Department of China, Qingdao 266033, China

**Keywords:** steam-cured concrete, slag, autogenous shrinkage, hydration model, shrinkage model

## Abstract

In order to better predict the development of shrinkage deformation of steam-cured concrete mixed with slag, a deformation-temperature-humidity integrated model test, a hydration heat test, and an elastic modulus test were performed. The effects of the steam-curing process and the content of slag on shrinkage deformation, hydration degree and elastic modulus of concrete were studied. The results indicate that during the steam-curing process, the concrete has an “expansion-shrinkage” pattern. After the steam curing, the deformation of concrete is dominated by drying shrinkage. After the addition of slag, the shrinkage deformation of steam-cured concrete is increased. The autogenous shrinkage increases by 0.5–12%, and the total shrinkage increases by 1.5–8% at 60 days. At the same time, slag reduces the hydration degree of steam-cured concrete and modulus of elasticity. A prediction model for the hydration degree of steam-cured concrete is established, which can be used to calculate the degree of hydration at any curing age. Based on the capillary tension generated by the capillary pores in concrete, an integrated model of autogenous shrinkage and total shrinkage is established with the relative humidity directly related to the water loss in the concrete as the driving parameter. Whether the shrinkage deformation is caused by hydration reaction or the external environment, this model can better predict the shrinkage deformation of steam-cured concrete.

## 1. Introduction

With the improvement of current standards for building industrialization, environmental protection and low-carbon concepts cannot be ignored. Traditional cast-in-place concrete technology has a series of shortcomings, such as environmental pollution, an unstable quality of concrete, high resource consumption and being labour-intensive [1]. However, prefabricated building components can not only improve the production efficiency and quality of concrete components, and free workers from heavy manual labour, but also better control the garbage and noise pollution generated on site [2,3]. At the same time, prefabricated construction technology has become an important and effective way to modernize the construction industry [4]. According to statistics, in the actual production of prefabricated building components, over 70% of precast concrete is cured by steam. Steam curing can not only accelerate the hydration of cementitious material to improve the microstructure of concrete but also increase the utilization of the rate of equipment, shorten the production cycle, reduce environmental impact and reduce product costs [5,6]. However, steam curing will also adversely affect the internal pore structure of concrete, easily causing concrete swelling and deformation, the delaying expansion of ettringite, and affecting the volume stability of steam-cured concrete [7,8]. Mineral admixtures such as fly ash and slag are important materials in concrete. Mineral admixtures can help reduce the production cost of steam-cured concrete and can improve some of the undesirable properties of steam-cured concrete. Slag is an important by-product in industrial production, and it has been widely used in the production of precast concrete [9,10,11,12].

Concrete is a typical hydrophilic material with a large number of tiny pores and cracks inside. when the environmental relative humidity is lower than the internal relative humidity of concrete, the moisture inside the concrete will evaporate and cause concrete to shrink. This kind of shrinkage is called drying shrinkage. Another type of shrinkage is caused by the consumption of water due to the hydration of cementitious material during the concrete setting and curing process at early ages and is called autogenous shrinkage. As is known, the shrinkage strain of concrete is positively correlated with the relative humidity inside concrete [13,14,15]. However, shrinkage of concrete is one of the major mechanisms leading to the formation of initial cracks in concrete. With the development of shrinkage, tensile stress caused by shrinkage will increase in concrete. When the tensile stress exceeds the tensile strength of concrete, the concrete will crack. The cracks caused by shrinkage will reduce the durability of concrete and reduce the safety, reliability and service life of buildings, especially for buildings with large surface area and long-term exposure to the external environment. In recent years, the autogenous shrinkage and drying shrinkage deformation of concrete have become popular research topics [16,17,18]. Most researchers study drying shrinkage and autogenous shrinkage, respectively. When buildings are exposed to the natural environment, autogenous shrinkage caused by cement hydration and drying shrinkage caused by water evaporation occur simultaneously. Shrinkage of concrete is the sum of autogenous shrinkage and drying shrinkage and ignoring any of them will have a greater impact on subsequent studies on shrinkage deformation of concrete. Therefore, a comprehensive study of the autogenous shrinkage and drying shrinkage of concrete mixed with slag is of great significance to the early cracking of concrete. The establishment of a shrinkage model that integrates the autogenous shrinkage and drying shrinkage of slag concrete is very important for predicting the early deformation development of slag concrete. If the model is successfully established, the deformation of the concrete mixed with slag can be predicted by only a few parameters, and the shrinkage and cracking of the concrete can be prevented in time according to the prediction results to avoid greater structural damage to the concrete. Shah et al. [19] used bagasse instead of cement to study the related properties of concrete, and established a model to predict the mechanical properties of concrete based on MEP, which provided a reference value for the study of the waste disposal crisis and construction industry simulation. In addition, a large number of current studies only focus on a certain parameter (such as age) and establish a simple shrinkage model with a single factor as a variable. This does not properly reflect the influence of the age, the cement hydration process, structural shape, size, cross-sectional position, and ambient temperature and humidity boundary conditions on shrinkage distribution and development. Zhang [20] established a micromechanical model for predicting the autogenous shrinkage and drying shrinkage of concrete. The prediction results of the model were compared with the experimental results of C30, C50, C80 concrete. It was found that there is a good consistency between them.

In this paper, the effects of the steam curing process and addition of slag on hydration, deformation, pore size distribution and elastic modulus of concrete were studied. Based on the experimental results, an integrated model of autogenous shrinkage and total shrinkage of steam-cured concrete based on changes of internal relative humidity and temperature will be established. The results are helpful for understanding and predicting the shrinkage and deformation of concrete after steam curing.

## 2. Experimental Program

### 2.1. Mixture Proportions of Concrete

Portland cement type of P.I 52.5 and slag type of S95 with chemical composition were used (Cf. Table 1). The specific surface area and loss on ignition are 450 m^2^/kg and 0.8%, respectively. The fine aggregate was river sand, and its fineness modulus was 2.6. The coarse aggregate was basalt, and its continuous gradation was 5–20 mm. The water reducing agent is a polycarboxylic acid-based superplasticizer with a water-reducing rate of 30%. The mixing water was tap water. The water-binder ratio of the cement paste is 0.32, and the mix proportion of the concrete segment of the typical coastal subway is shown in Table 2. The amount of superplasticizer depends on the workability of concrete, around 200 mm according to the slump test. By adding the same quality of slag to replace the same quality of cement, the range of slag dosage was 15–50% of the total amount of cementitious material. The steaming system is shown in Figure 1. First, the concrete is cured for 2 h at a curing temperature of 35 °C, then uniformly heated to 60 °C within 2 h, cured for 2 h at a constant temperature of 60 °C, and then uniformly cooled to 35 °C within 2 h. After the steam curing is over, the concrete specimens used for elastic modulus testing are put into the standard curing room (temperature (T) = 20 ± 2 °C, relative humidity (RH) ≥ 95%) for curing, and after curing to the corresponding age, they are taken out for testing. After the steam curing is completed, the concrete specimen used for the deformation test is placed in a constant temperature and humidity room (T = 20 ± 2 °C, RH = 50 ± 2%) for curing.

### 2.2. Experimental Tests

#### 2.2.1. Hydration Heat

The TAMAir eight-channel isothermal calorimeter (TA Instruments -Waters China Limited, Hong Kong, China) was used to test the heat of hydration of the pure pulp samples S0 and S25 at 20 °C and 50 °C. The data collection interval was 30 s, and the test time was 7 days. This uses water as the reference group. It puts the weighed sample into an ampoule and weighs 4 g of the gelling material sample. They are stirred at low speed in the ampoule.

#### 2.2.2. Capillary Negative Pressure

The capillary negative pressure test system is based on physics and combined with the Laplace equation to develop a comprehensive test system for capillary negative pressure. It can realize real-time unattended monitoring of the capillary negative pressure of cement-based materials by managers [21,22,23,24]. The system consists of a pressure transmitter, a microporous ceramic head, a tetrafluoroethylene water storage pipe, a negative pressure test device and a data acquisition and storage device, as shown in Figure 2.

The test piece uses a steel mold of 100 mm × 100 mm × 400 mm. Before the start of the test, the sensor device and the water potential probe are exhausted first, and then the microporous ceramic head is buried in the sample to be tested, and the initial value of the test is 0 ± 0.5 kPa through change. After the test piece is poured, it is moved into the steam curing box. The test collection instrument will automatically record the capillary negative pressure value of the entire process, and the collection frequency is 0.5–2 min/time.

#### 2.2.3. Internal Relative Humidity Test

A vibrating wire strain gauge with a range of ±1500 με is used to measure the deformation of concrete. Its accuracy is 1 με, the measuring gauge length is 150 mm, and the applicable ambient temperature is −20–125 °C. The strain gauge can automatically perform temperature compensation. The fresh concrete is poured into a steel mould of 100 mm × 100 mm × 400 mm. The strain gauge is pre-embedded in the centre of the test piece, and the temperature and humidity sensor is inserted at half the height of the concrete specimen. It is necessary to calibrate the temperature and humidity sensor with the salt solution before the test. Table 3 shows the theoretical value of salt solution RH [25], Figure 3 shows the RH calibration value. It can be seen from the figure that the linear correlation coefficient between the RH value measured by the temperature and humidity sensor and the theoretical value is 0.9999, which shows a high degree of agreement. Then the concrete is poured. After the concrete is vibrated and compacted, it is moved with the mould into the steam curing box for steam curing immediately. At the same time, the data acquisition system is turned on. During the steaming process, the collection frequency is every 10 min. The concrete was demoulded in about 9 h, and part of the specimen was sealed with aluminium foil tape to measure the autogenous shrinkage deformation, and part of the specimen was left unsealed with two long sides to measure the total shrinkage deformation (expressed by semi-sealing). Then the specimen is moved into a constant temperature and humidity room (T = 20 ± 2 °C, RH = 50 ± 2%) to continue curing, and the collection frequency is every 30 min. The test setup is shown in Figure 4.

#### 2.2.4. Porometry Assessment

Low-field nuclear magnetic resonance (LF-NMR) is a new technique for rapid measurement of rock physical parameters. It has the advantages of fast measurement, simple operation, no damage to the core, no pollution, and quick acquisition of reservoir physical parameters, which provides a novel way to evaluate the pore structure of the reservoir. This paper uses this technology to obtain the pore distribution characteristics of the cement paste after steam curing. Because of the short setback time of this technology, the relaxation of free water can be ignored [26,27]. Assuming that the pores are homogeneous and cylindrical, the pore diameter d in the cement paste can be calculated according to Equation (1) [28]:(1)$$d=4\rho_{2}T_{2}$$
where *ρ*_2_ is the surface relaxation rate, which is 12 nm/ms [29]; *T*_2_ is the lateral relaxation time of water molecules in the pore, ms.

The test selects cement paste samples with ages of 8 h, 7 days and 28 days for testing. Before the start of the test, the sample was crushed, then placed in absolute ethanol for 7 days to end the hydration process, and finally dried in an oven at 50 ± 2 °C for 3 days before the NMR test. During the NMR test, the magnetic field intensity of the instrument was set to 0.42 T, the magnet frequency was set to 18 MHZ, and the magnet temperature was maintained at 32 ± 0.02 °C. The temperature in the laboratory was controlled at 20 ± 1 °C through air conditioning and other equipment, and the sample was placed in a cylindrical glass tube with a diameter of 25 mm and a length of 200 mm for testing. The standard oil sample was calibrated by the free induction attenuation sequence to obtain the frequency deviation (O1) and 90° pulse width (P1) of the RF signal. The transverse relaxation time (T2) of the sample was obtained by the Can–Purcell–Meiboom–Gill pulse sequence (π/2–τ–NECH π, half echo time τ is 180 s, NECH is 500), and finally the pore size inside the cement paste was obtained.

#### 2.2.5. Dynamic Elastic Modulus

The modulus of elasticity reflects the relationship between the stress and strain on concrete, and it is an important parameter for establishing a concrete shrinkage model [30]. Research shows that the static elastic modulus and dynamic elastic modulus of concrete can be converted by Equation (2) [31,32]. The test results of Xu Guodong [32,33,34] et al. were fitted using Equation (2), and the parameters k and c were taken as 0.739 and 0.021, respectively, as shown in Figure 5. Therefore, we used an ultrasonic detector to test the dynamic elastic modulus of concrete at different ages [35]. The ultrasonic transmitting end and receiving end were placed on both sides of the concrete test block to collect the ultrasonic pulse signal penetrating the concrete and capture the data on the video display to record the test results. The parameters such as the time and wave speed of the ultrasonic wave passing through the concrete were analyzed to calculate the dynamic elastic modulus of the concrete, as shown in Equation (3). Then it was converted to static elastic modulus according to Equation (2). The size of the concrete specimen was 100 mm × 100 mm × 100 mm. After steam curing, the specimen was moved into a standard curing room. When the concrete had cured for 8 hours, 1 day, 3 days, 7 days, 14 days, 28 days and 56 days, it was taken out and tested.
(2)$$E_{S}=E_{d}\left(1-Ke^{-cE_{d}}\right)$$
(3)$$E_{d}=\frac{\left(1+\upsilon \right)\left(1-2\upsilon \right)\rho L^{2}}{\left(1-\upsilon \right)t^{2}}$$
where *E_d_* is the dynamic elastic modulus; *E_s_* is the static elastic modulus; *L* is the length of the specimen, m; *ρ* is the concrete test piece density, kg/m^3^; *υ* is Poisson’s ratio; *t* is the age of the test piece in full immersion solution; *K*, *c* are fitting parameters.

## 3. Experimental Results

### 3.1. Hydration Heat of Cement Paste with and without Slag at Different Temperature

Figure 6 shows the hydration rate and cumulative heat release curve of cement paste with and without slag at different temperature. The hydration process of cement paste is mainly divided into the induction period, acceleration period and deceleration period. Among them, the induction period refers to the formation of high-Si hydration products within 2–3 h after the cement and water are mixed, which hinders the further hydration of C_3_S and slows the hydration rate, and the cement still maintains good plasticity. It can be seen from Figure 6a that in the induction period, slag has no obvious effect on the induction period of cement paste at 20 °C. However, the specimen did not show obvious induction time at 50 °C. In the acceleration period, the addition of slag delayed the acceleration period of cement paste. Compared with 50 °C, when the temperature is 20 °C, the activity of the slag is lower, which makes the self-hydration reaction of the slag slower. In addition, slag adsorbs Ca^2+^ in the pore solution and releases more SiO_4_^4−^, which reduces the Ca/Si ratio in the pore solution, so unstable C–S–H is formed [36,37]. It takes a certain time for this unstable C–S–H to transform into stable C–S–H, which delays the acceleration period. Slag will reduce the exothermic temperature peak of cement paste, which is mainly caused by the hydration reaction of C_3_S and C_2_S [33]. From the overall hydration curve, slag reduces the hydration rate. In addition, the high temperature promotes the hydration rate during the acceleration phase and the high temperature during the deceleration period reduces the hydration rate. This is mainly because the high temperature in the early stage promotes the chemical reaction of slag to promote hydration. With the progress of hydration, an enormous amount of C–S–H gel is generated to wrap the cement and slag particles. In this way, the continuation of hydration is hindered.

It can be seen from Figure 6b that under the condition of 20 °C, the addition of slag reduces the cumulative heat release. The main reason is that the hydration reaction of slag was slow at room temperature, so the cement pastes mainly rely on the cement hydration reaction to release heat at the early stage. However, under the condition of 50 °C, slag increases the cumulative heat release. This is because high-temperature conditions promote the hydration reaction of cement and form more Ca(OH)_2_. Ca(OH)_2_ plays the role of an alkali activator. Under the action of the alkali activator, slag will react with the Ca(OH)_2_ in the pore solution. Meanwhile, slag will hydrate under high temperature. The above processes result in the release of more heat.

### 3.2. Deformation of Steam-Cured Slag Concrete

#### 3.2.1. Deformation of Slag Concrete during Steam Curing Stage

The development of deformation during the steam curing stage is caused by a combination of many factors, including the thermal expansion of each phase of concrete, chemical shrinkage caused by hydration, and autogenous shrinkage caused by lower internal humidity. Relationship between deformation of concrete and the internal temperature and relative humidity during the steam curing stage is shown in Figure 7. In the pre-curing stage (0–2 h), the internal capillary structure of the concrete sample has not yet formed. As shown in Figure 8, the sample cannot form a stable hole. The hydration products formed inside the system have not overlapped each other to form a self-supporting stable structure, and the capillary grid similar to the solid phase structure cannot be formed, so the system cannot stabilize the pores. The cement paste tends to collapse, causing lateral expansion. At the same time, the interior of the samples expands due to heat, causing the overall thermal expansion of the sample. During the heating period (2–4 h), the temperature continues to rise to promote the hydration process of cementitious material. The hydration products overlap each other to form a self-supporting structure inside the system, and a capillary grid structure system similar to a solid phase structure form. The thermal expansion coefficient of early age concrete is much greater than that of steel moulds [38], and the modulus of elasticity is small, resulting in greater temperature deformation of concrete than steel moulds. The steel mould has a constraining effect on the concrete. However, the humidity inside concrete does not decrease and the autogenous shrinkage is small, so the macro-volume deformation of concrete manifests as expansion and the expansion peak appears. Therefore, through this analysis, it is recommended that in the subsequent tests, sufficient space should be left in the steel mold to allow the concrete to deform freely, so as to reduce the influence of the steel mold on the deformation of the concrete. After the expansion peak, the deformation curve decreases, and the relative humidity curve decreases at the same time. In addition, at this stage, compared with concrete just with cement, other mix proportion of concrete with cement and slag not only delay the appearance of the expansion peak but also increase the peak value of expansion. This is mainly because the high temperature promotes the hydration of cement and generates more alkaline substances. Under the alkaline conditions, OH^−^ can destroy the Ca–O bond, Al–O bond and Si–O bond in slag, which promotes the hydration of slag. Therefore, the amount of Ca(OH)_2_, Mg(OH)_2_ and ettringite in the hydration product increases faster than concrete just with cement, which increases the expansion level [33]. In the constant temperature stage (4–6 h), as the relative humidity decreases, the internal capillary network structure transforms into an unsaturated state. Further hydration of cement is accompanied by chemical shrinkage to form the meniscus of the pore network in the concrete. There is a pressure difference between the vapour and liquid phases across the meniscus, causing the system to generate capillary negative pressure, and then shrinkage deformation begins. At the same time, the internal temperature of the concrete is almost stable at this stage. Therefore, the decrease in relative humidity inside concrete at this time is the main factor that causes concrete to shrink. In the cooling stage (6–8 h), the strain value of the concrete sample is stable, but at the end of the cooling stage, a slight expansion phenomenon appears, which is considered to be caused by the delayed ettringite caused by the temperature drop [39,40].

#### 3.2.2. Deformation of Slag Concrete after Stream Curing

As is known, the precast concrete samples are always steam cured in the factory, and then transported to the construction site. The steam curing process completed in the factory does not affect the application of the precast concrete samples in the construction site, so it is reasonable to use the time of demould as the zero point of shrinkage.

Figure 9 shows the correlation diagram between the autogenous shrinkage, total shrinkage and relative humidity of concrete after demold. Obviously, the addition of slag can increase the autogenous shrinkage of stream cured concrete after demould, and as the amount of slag added increases, the increase in autogenous shrinkage of stream cured concrete increases. Compared with S0, the increase of S15, S25 and S50 at 60 days was 0.9%, 7.1% and 11.7%, respectively. Slag also has an increasing effect on the total shrinkage. Compared with S0, the increase of total shrinkage of stream cured concrete at 60 days was 1.5%, 6.4% and 7.3%, respectively. The major reasons are as follows: (1) The filling and dilution effect of slag promotes the hydration of cement. At the same time, after steam curing, high temperature promotes the pozzolanic effect of slag, which significantly increases autogenous shrinkage of concrete. (2) Slag contains a lot of SiO_2_ and Al_2_O_3_, the autogenous shrinkage will increase significantly. Mainly due to the active SiO_2_ and Al_2_O_3_, the hydration products are more complicated, including C–S–H gel (calcium–silicate–hydrate), C–A–S–H gel (calcium sulphoaluminate), C–A–C–H gel, hydrotalcite-like material, and C–A–H (calcium aluminate hydrate). Compared with hydration products of cement, these hydration products contain more bounded water, which affects the autogenous shrinkage of concrete [41,42,43]. (3) Compared with slag, cement contains more SO_4_^2−^ and generates more ettringite, which makes the concrete expand more, so it plays a supplementary role in autogenous shrinkage. Therefore, in the actual project, we tried not to mix mineral powder alone, and it could be mixed with fly ash. The internal shrinkage of the concrete specimen increases with the decrease of relative humidity. The shrinkage curve and the relative humidity curve show good correlation and synchronization during the test period. The relationship between autogenous shrinkage and relative humidity is mainly manifested in the fact that when cement undergoes hydration, it absorbs a large amount of free water inside concrete to form voids, which causes the internal balance of unbound water to be lost, so the relative humidity continues to drop. For concrete, which is in a completely sealed state, the relative humidity of concrete continues to decrease, resulting in the presence of gas-phase in the pores, so that the water vapour in the pores is no longer saturated. As the cement hydration reaction continues to intensify, according to the principle of minimum energy, water must migrate from large pores to small pores driven by capillary action, and form a meniscus at the largest pore to reach a thermodynamically stable state. At a certain moment of the hydration reaction, there is a critical radius r_0_, and there is a meniscus in the capillary pores with a diameter of r = r_0_, which then generates capillary stress. This will produce negative pressure on concrete, which will eventually cause concrete to shrink automatically. The drying shrinkage of concrete is mainly manifested in two aspects: on the one hand, the hydration process consumes internal water and reduces the relative humidity of concrete. On the other hand, the relative humidity inside the concrete is constantly changing after exposure to the environment with different temperature and relative humidity, which leads to the loss of moisture inside the concrete to the environment. Through the above analysis, it can be found that both autogenous shrinkage and drying shrinkage are caused by the drop in relative humidity of concrete. Therefore, from the Kelvin equation [44,45], as the relative humidity continues to decrease, the radius of the capillary meniscus becomes smaller and the capillary negative pressure increases. The greater the tensile stress generated by the capillary wall under the negative pressure of the capillary pore, the greater the shrinkage of the concrete.

In addition, it can be seen from Figure 9 that the shrinkage value of the semi-sealed sample is almost twice that of the fully sealed sample at 3 days after steam curing. At 60 days, the drying shrinkage accounted for over 65% of the total shrinkage. Similarly, the total shrinkage of the concrete specimens corresponds to a lower relative humidity than the autogenous shrinkage. This means that drying shrinkage is the dominant form of shrinkage. This is mainly due to two aspects: on the one hand, the autogenous shrinkage under semi-sealed condition will also occur. On the other hand, the internal humidity of concrete will spread outward because of the humidity difference between the concrete and the environment. Under these two kinds of action, there is a humidity gradient in concrete, which leads to uneven shrinkage deformation. The surface shrinkage is large while the internal shrinkage is small, which causes dry shrinkage stress. When the dry shrinkage stress is greater than the tensile strength of concrete, it will cause surface cracking of the concrete, which provides a convenient channel for water loss. With the extension of exposure time, the water continues to be lost from the concrete. According to Kelvin’s law, the radius of the meniscus of the capillary keeps decreasing, and the negative pressure of the capillary keeps increasing, which leads to an increase in the drying shrinkage. Because the concrete under full sealing condition reaches a high degree of hydration after steam curing, the degree of continuous hydration of cementitious materials is low. Therefore, the amount of autogenous shrinkage is gradually reduced. Moreover, according to the disjoining pressure theory [46], in the initial stage, the solid surfaces of concrete are very close and the relative humidity is high. In order to increase the thickness of the adsorbed water molecular layer, the adsorbed water tends to separate the two solid surfaces, which makes the two solid surfaces bear the disjoining pressure. When concrete changes from the saturated state to the unsaturated state, the disjoining pressure begins to decrease, which makes the two solid surfaces closer together, resulting in shrinkage deformation. At the same time, due to the lower relative humidity under semi sealing condition, the disjoining pressure is lower. Finally, the shrinkage of concrete is greater. Therefore, in order to avoid greater drying shrinkage, it is necessary to avoid exposing the test piece to the external environment for a long time.

### 3.3. Pore Size Distribution of Steam-Cured Slag Concrete

The shrinkage model in Section 4.2 is based on the capillary pressure theory [44,45]. The pore structure of the steam-cured cementitious material is very important for the analysis of the shrinkage mechanism of steam-cured concrete. According to the Kelvin–Laplace equation of the capillary pressure theory, the diameter of the capillary pores is inversely proportional to the shrinkage tensile stress. The larger the diameter of the capillary pores, the smaller the shrinkage tensile stress.

Figure 10 shows the cumulative pore volume of S0 and S25 at 8 h, 14 days and 28 days. Obviously, the cumulative pore volume of S0 and S25 decreases as the pore diameter increases. The overall trend showed a slow decline and then a rapid decline, and finally approached a flat trend. Compared with S0, S25 has a higher cumulative pore volume at all ages. Figure 11 and Figure 12 show the change of the pore size distribution of S0 and S25 with time. It can be seen from the figures that all curves contain two peaks. The higher peak is the gel pores formed by the hydration product of calcium silicate hydrate (C–S–H) gel, the lower peak is the capillary pores formed by the mixed water that has not participated in the hydration reaction [47,48,49]. In addition, after the addition of slag, the gel pores tend to move to the right and the area of the gel pores increases. This is mainly due to the increase of expansion hydration products because of the early curing environment and the nature of slag. At the same time, the influence of thermal expansion and internal stress makes the pore structure in concrete rough. With the increase of curing time, the pore size decreases. This trend is due to more hydration products and the higher density of hydrated products. The above conclusions are consistent with the results of Supit et al. [50,51].

Figure 13 shows the change in the most probable pore diameter of S0 and S25. Obviously, The most probable pore size of S0 and S25 is less than 20 nm, and they are all harmless pores. At the same age, the most probable pore diameter of the samples increases by 23–32% because of the addition of slag. At 8 h, 14 days and 28 days, the most probable pore diameter of S25 is 3.11 nm, 3.74 nm and 3.17 nm larger than that of S0, respectively.

### 3.4. Elastic Modulus of Steam-Cured Slag Concrete

Figure 14 shows the static elastic modulus obtained by converting the measured dynamic elastic modulus of steam-cured concrete according to Equation (2). The elastic modulus of steam-cured concrete gradually decreases with the increase of slag content during the early stage, but the elastic modulus of steam-cured concrete is basically the same during the latter stage. This is mainly because the incorporation of slag reduces the amount of cement. As mentioned above, the pozzolanic reaction of slag with Ca(OH)_2_ is behind the hydration reaction of cement. With the increase of curing age, the high-calcium components in slag continuously decomposes to pore solution, which promotes the generation of Aft which is a crystalline hydrated calcium sulfoaluminate produced by the combination of cement hydration product C–A–H (calcium aluminate hydrate) and sulfate ions, and gradually increases the elastic modulus. At the same time, the components in slag react with Ca(OH)_2_ to form C–S–H, which reduces the content of Ca(OH)_2_ at the interface between hydrate and coarse aggregate. As a result, the interface transition zone is improved [52]. This improves the quality of calcium silicate hydrate gel and then improves the elastic modulus.

BS EN [53] shows the development of elastic modulus of concrete conforms to the law of Equation (4), where *a* and *b* are constants and their values are shown by the fitting curves. The results in Figure 14 are fitted by Equation (4). The correlation coefficient is greater than 0.99. Therefore, the elastic modulus of steam-cured slag concrete can be calculated according to the fitting equations.
(4)$$\;y=ax^{b}$$
where *a*, *b* are fitting parameters.

## 4. Discussion

### 4.1. Equivalent Hydration

The temperature is constantly changing during steam curing. In order to analyze the influence of temperature on the hydration of cementitious material, the concept of hydration degree is introduced. The hydration process of cementitious materials can be quantified by the degree of hydration, and the degree of hydration is an important parameter of the shrinkage model of concrete. The degree of hydration *α*(*t*) can be calculated based on the cumulative heat release *Q*(*t*) and the heat release *Q*(*u*) to achieve complete hydration, as shown in Equation (5). The ultimate heat release *Q*(*u*) can be determined according to the Q-FHP model as shown in Equation (6), where *Q*(*u*) = a1 + a2 [31,54,55]. The cumulative heat release of the cement paste in 7 days at 20 °C and 50 °C in Figure 6b is fitted by Equation (6) as shown in Figure 15. From Table 4, it can be seen that *Q*(*u*) of the samples with the same mix proportion is almost the same at different temperature. This is because *Q*(*u*) is just related to the mix proportion of the samples. The average *Q*(*u*) of S0 and S25 is 353.61 J/g and 446.27 J/g, respectively. Furthermore, the change curve of the hydration degree of the cement paste at 20 °C and 50 °C is shown in Figure 16a.
(5)$$\alpha \left(t\right)=\frac{Q\left(t\right)}{Q\left(u\right)}$$
(6)$$Q\left(t\right){=a}_{1}\cdot e^{-{\left(\frac{\tau_{1}}{t}\right)}^{\beta_{1}}}+a_{2}\cdot e^{-{\left(\frac{\tau_{2}}{t}\right)}^{\beta_{2}}}$$
where a_1_, a_2_, *τ*_1_, *τ*_2_, *β*_1_, and *β*_2_ are parameters.

The steam-curing process is a temperature-changing process. In order to eliminate the influence of temperature history on the analysis of the hydration degree of cementitious material, researchers have established the equivalent age equation, as shown in Equation (7) [54]. Through Equation (7), the equivalent age between different curing temperatures can be calculated.
(7)$$t_{e}\left(T\right)\;=\;\overset{t}{\underset{0}{\sum }}\exp\left[-\frac{E_{a}}{R}\left(\frac{1}{T}-\frac{1}{T_{r}}\right)\right]\times \Delta t$$
where *T_r_* is the reference temperature, *K*, taking 293 K; *E_a_* is the activation energy of the material, kJ/mol; *R* is the gas constant, J/mol∙K, taking 8.314 J/mol∙K; *T* is the average temperature in Δ*t*, *K*.

The apparent activation energy *Ea* in Equation (7) is the only parameter that reflects the sensitivity of concrete to temperature. Wade et al. [56,57] found that the activation energy was 35.3 kJ/mol–50.6 kJ/mol. After trial calculations, when *Ea* of S0 and S25 is 36 kJ/mol and 47 kJ/mol, respectively, compared with the measured hydration degree, the calculated hydration degree based on equivalent age has the highest accuracy, as shown in Figure 16a. In addition, the calculated hydration degree and equivalent age obtained in Figure 16a are substituted into the hydration degree model [58], as shown in Equation (8).
(8)$$\alpha \;=\;\exp{\left(-\left[\ln(1+\frac{t_{e}}{a}\right)\right]}^{b})$$
where *t_e_* is the equivalent age; *a* and *b* are constants. Parameters *a* and *b* of S0 and S25 are, respectively, 0.67, −2.25 and 1.35, −1.56. Then the hydration degree of the cementitious material at any age can be calculated.

In order to study the effect of steam curing on the hydration process of concrete, the establishment and verification of subsequent shrinkage models, the internal temperature change of concrete was measured as shown in Figure 17. According to Equation (7), the hydration degree of the cementitious material at a constant temperature within 7 days is converted to the hydration degree under variable temperature. The hydration degree after 7 days is calculated according to Equation (8), as shown in Figure 16b. Clearly, the hydration degree of the cement paste is reduced because of the addition of slag. The gap gradually narrows. The main reason is that the content of cement that reacts with water will be reduced by the addition of slag. Then the generation of CH is reduced. As a result, the pozzolanic effect of slag cannot be fully exerted, thereby reducing the hydration degree [59]. At the same time, while the high temperature in the steam curing stage accelerates the formation of hydration products, sufficient water enters concrete to disperse the hydration products, which is beneficial to the continuous hydration of the cement [60]. Furthermore, the final hydration degree of cement paste with the same mixing proportion under different conditions is the same, which is determined by the water–binder ratio.

### 4.2. Relationship between Shrinkage and Relative Humidity of Steam-Cured Concrete after Stream Curing

In this section, the change of relative humidity in concrete is used as the internal cause, and an integrated model of autogenous shrinkage and total shrinkage of steam-cured concrete is established based on elastic modulus, hydration degree and pore diameter. Since the model defaults to the zero time as the starting value, the calculated value of the model should be compared with the test results after demould to verify the applicability of the model. According to the self-consistent method, the shrinkage strain caused by the internal tensile stress $\sigma_{cap}$ is shown in Equation (9) [61,62]:(9)$$\varepsilon_{H}\;=\;\frac{S_{W}\sigma_{cap}}{3}\left(\frac{1}{K_{T}}-\frac{1}{K_{S}}\right)$$
where $S_{W}$ is the saturation coefficient; $\sigma_{cap}$ is the pore stress; $K_{T}$ is the bulk modulus of elasticity, GPa; $K_{S}$ is the bulk modulus of elasticity without pores, 44 GPa. The bulk modulus of elasticity is shown in Equation (10) [63]:(10)$$K_{T}\;=\;\frac{E}{3\left(1-2\mu \right)}\;$$
where *E* is the elastic modulus; μ is the Poisson’s ratio, 0.2.

The saturation coefficient $S_{W}$ of pure cement and cementitious material mixed with slag is calculated according to Equations (11) and (12), respectively [41,64]. *V_ew_* and *V_p_* are functions of the hydration degree.
(11)$$S_{W}\;=\;\frac{V_{\mathrm{ew}}\left(\alpha \right)}{V_{p}\left(\alpha \right)}\;=\;\frac{p-0.7\left(1-p\right)\alpha }{p-0.5\left(1-p\right)\alpha }\;$$
(12)$$S_{W}\;=\;\frac{V_{ew}\left(\alpha \right)}{V_{p}\left(\alpha \right)}\;=\;\frac{P-\left(0.72+0.37\left(S/C\right)\right)\left(1-P\right)K\alpha }{P-\left(0.52-0.11\left(S/C\right)\right)\left(1-P\right)K\alpha }$$
where *V_ew_* is the vaporizable water in the hardened cement paste; *V_p_* is the total pore volume; *P* is the water-cement ratio function; *K* is the ratio function.

*P* of pure cement and cementitious material mixed with slag is calculated according to Equations (13) and (14), respectively.
(13)$$P\;=\;\frac{W/C}{W/C+\rho_{w}/\rho_{c}}$$
(14)$$P\;=\;\frac{\frac{W}{C}}{\frac{W}{C}+\frac{\rho_{W}}{\rho_{C}}+\frac{\rho_{W}}{\rho_{S}}\times \frac{S}{C}}$$
where ${}_{C},{\;}_{W},{\;}_{S},{\;}_{C}$, ${}_{W}$, and ${}_{S}$ is the density of cement, water and slag respectively, taking 3120 kg/m^3^, 1000 kg/m^3^ and 2850 kg/m^3^, respectively; *W/C* is the mass fraction ratio of each component.

*K* of the cementitious material mixed with slag is calculated according to Equation (15):(15)$$K\;=\;\frac{1}{1+0.65\frac{S}{C}}$$

According to the Kelvin–Laplace equation [41], The steam-cured concrete shrinkage model is shown in Equation (16).
(16)$$\varepsilon_{H}\;=\;-\frac{S_{W}RT}{3V_{m}}\left(\frac{1}{K_{T}}-\frac{1}{K_{S}}\right)\ln\left(RH\right)$$
where *RH* is the relative humidity, %; *T* is the Kelvin temperature, K; $V_{m}$ is the molar volume of water, 1.8 × 10 m^3^/mol.

Except for the deformation caused by the change of relative humidity, the temperature deformation caused by the change of temperature is another important part of the early deformation of concrete. In order to accurately predict the overall deformation, the deformation caused by temperature must be considered. The total deformation *ε* measured by the test should be composed of the deformation *ε_h_* caused by the change of relative humidity and the deformation *ε_t_* caused by the change of temperature, as shown in Equation (17).
(17)$$\varepsilon \;=\;\varepsilon_{H}+\varepsilon_{T}$$

The early temperature deformation is mainly affected by the thermal expansion coefficient *β_t_* and temperature *T* of concrete. Figure 17 shows the temperature-time curve. The estimation of the early thermal expansion coefficient is shown in Equation (18) [20].
(18)$$\beta_{t}\;=\;C\cdot \exp(-\gamma \cdot t_{e})+\beta_{0}$$
where *t_e_* is the equivalent age of the cement hydration process; *C*, *λ* and *β*_0_ are constants, the values of which are 132, 0.3 and 7, respectively.

Figure 18 shows the comparison of shrinkage test values of S0 and S25 and calculation results. Under the sealed condition, the autogenous shrinkage model can capture the development characteristics of the shrinkage strain of concrete well from the beginning of the setting. The curve initially rises to a certain stage at a relatively fast speed, and then gradually decreases to a relatively slow stage. Under the semi-sealed condition, the anastomosis was good in the early stage, and the calculated value of the model after 28 days was slightly lower than the experimental value. The primary reason is that when the concrete is continuously exposed to the external environment, after the free water in the pores is consumed, the adsorbed water will also be consumed. The adsorbed water is maintained by the static tension in the capillary. When the adsorbed water is lost, the concrete shrinks and the shrinkage caused by the loss of absorbed water is much larger than the shrinkage caused by the loss of free water. At the same time, long-term exposure will also consume the free water in the hydrated calcium silicate gel and the interlayer water between the hydrated calcium silicate gel, which will further increase the shrinkage of the concrete. However, the temperature and humidity sensor is not sensitive to these parts of water, resulting in a relatively high measured relative humidity, which in turn makes the value calculated by the shrinkage model smaller.

## 5. Conclusions

In this paper, the development characteristics of shrinkage deformation of steam-cured concrete with and without slag are studied, and an integrated predictive model of autogenous shrinkage and total shrinkage is established. According to the research results, the following conclusions are drawn:During the steam curing process, the concrete shows an “expansion-shrinkage” pattern. After the mould is removed, the relative humidity and concrete deformation have good consistency, no matter in the fully-sealed or semi-sealed conditions. Shrinkage of concrete increases continuously with the decrease of the relative humidity. After steam curing, the main shrinkage of concrete is drying shrinkage. At 60 days, drying shrinkage accounts for over 65% of the total shrinkage. Compared with pure cement concrete, the addition of slag increases the autogenous shrinkage of concrete by 0.5–12%, and the total shrinkage increases by 1.5–8%.Compared with pure cement steam-cured concrete, the addition of slag reduces the hydration degree and elastic modulus of concrete. At the same time, the addition of slag increases the most probable pore diameter of cement paste, evacuates the hardened structure, and makes the pore distribution more reasonable.The hydration degree prediction development model of steam-cured concrete is established, and the hydration degree at any age can be calculated.An integrated model of concrete autogenous shrinkage and total shrinkage under steam curing conditions based on internal factors of the change of relative humidity was established. Whether shrinkage deformation is caused by cement hydration or drying, this model can better predict the deformation of concrete. In particular, the shrinkage of early age concrete is predicted. The deformation of early-age concrete is much more complicated than that of later mature concrete. This stage cannot be ignored.


## Figures and Tables

**Figure 1 materials-14-01647-f001:**
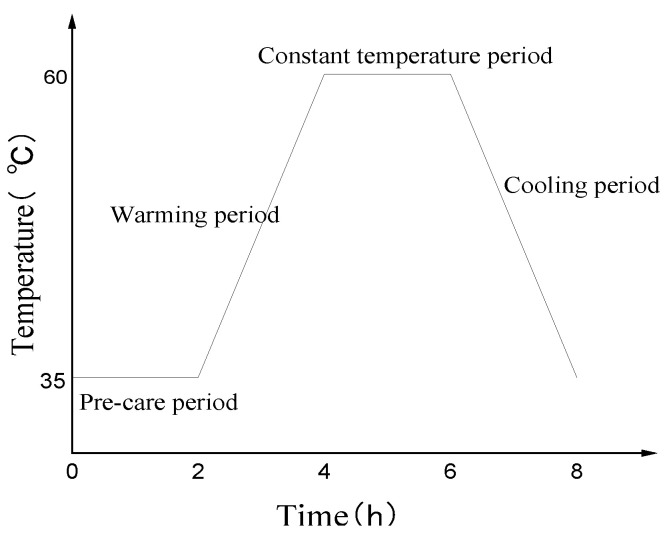
Steam-curing system.

**Figure 2 materials-14-01647-f002:**
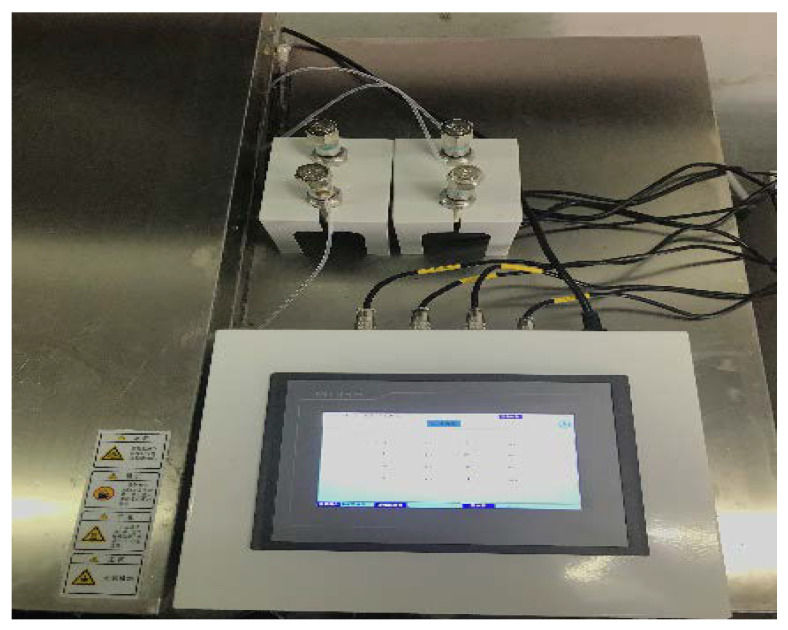
Capillary negative pressure instrument.

**Figure 3 materials-14-01647-f003:**
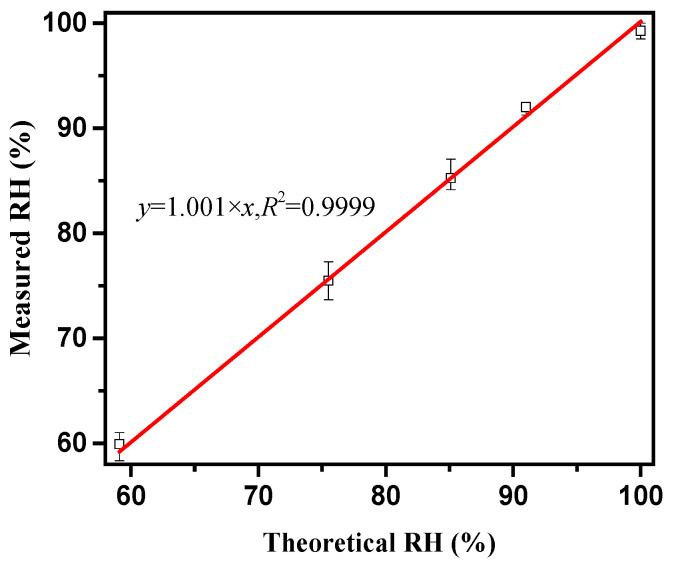
Temperature and humidity sensor calibration curve.

**Figure 4 materials-14-01647-f004:**
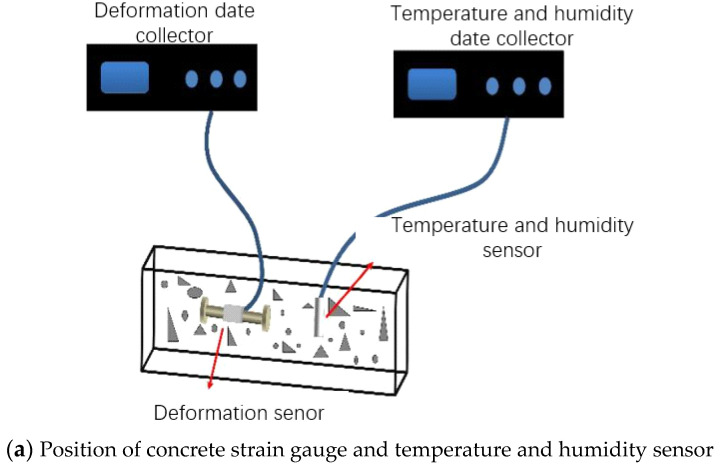
Test device of autogenous shrinkage, temperature and relative humidity (RH) of concrete.

**Figure 5 materials-14-01647-f005:**
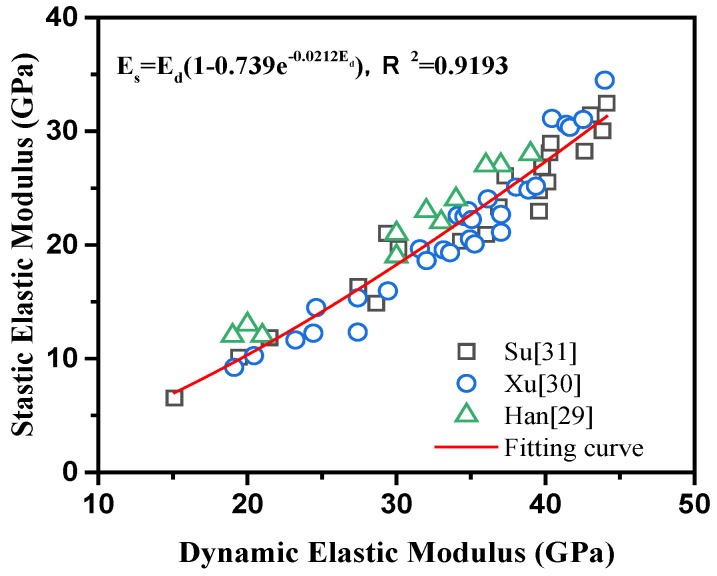
The relationship between static elastic modulus and dynamic elastic modulus of concrete.

**Figure 6 materials-14-01647-f006:**
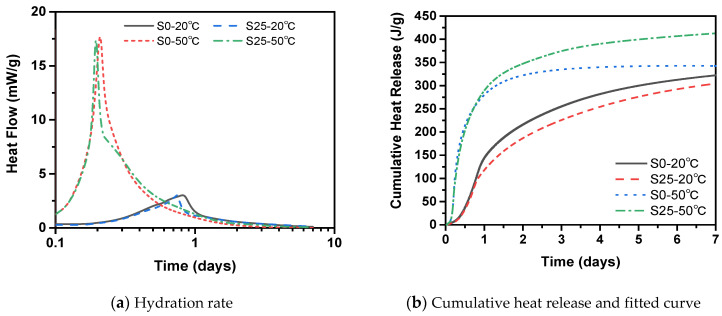
Hydration rate and cumulative heat release of cement paste with and without slag at different temperature.

**Figure 7 materials-14-01647-f007:**
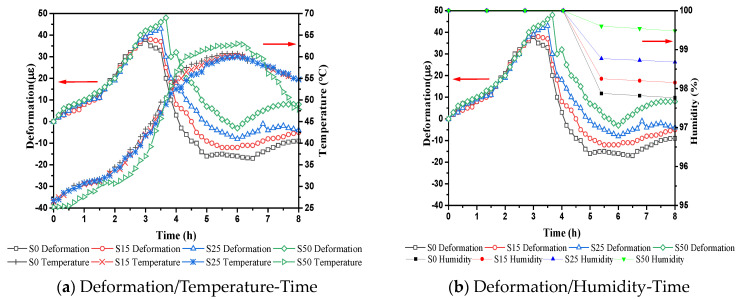
Relationship between deformation of concrete and the internal temperature and relative humidity during the steam curing stage.

**Figure 8 materials-14-01647-f008:**
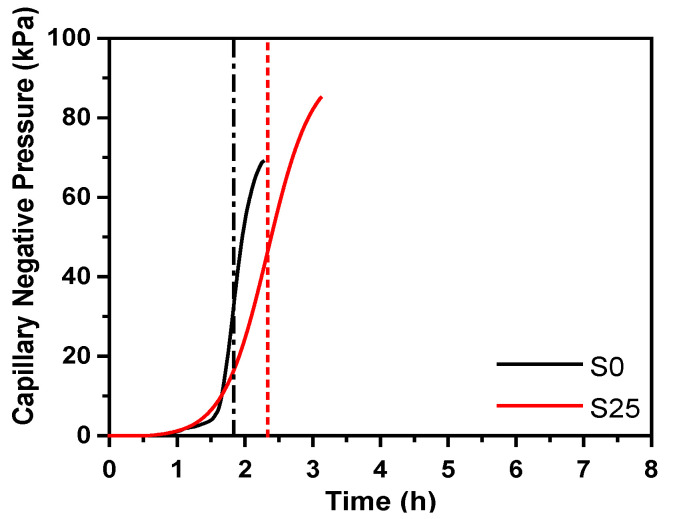
The development curve of capillary pressure inside concrete (the dotted line corresponds to the final setting time of concrete).

**Figure 9 materials-14-01647-f009:**
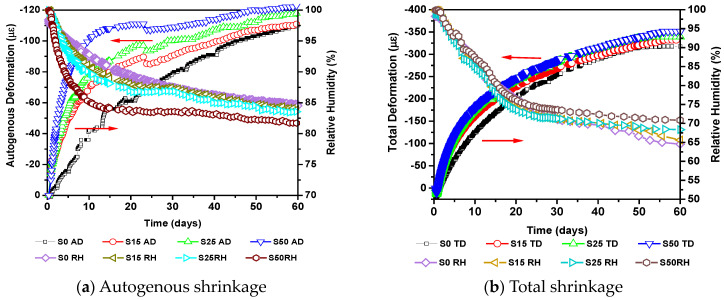
Deformation of slag concrete after stream curing.

**Figure 10 materials-14-01647-f010:**
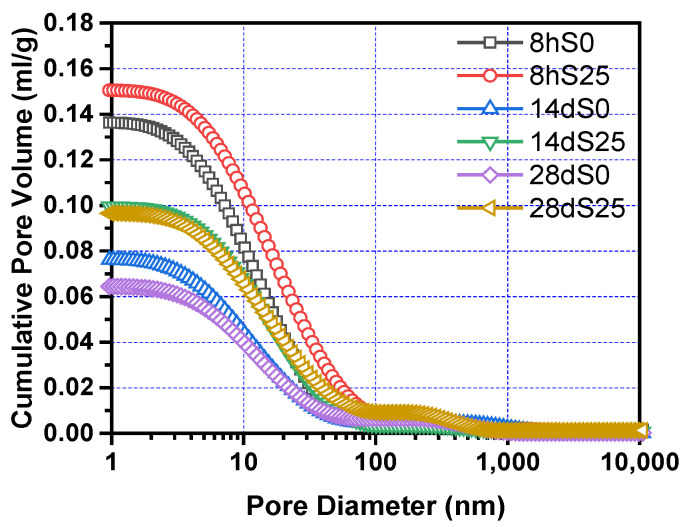
Cumulative pore volume of S0 and S25 at different times.

**Figure 11 materials-14-01647-f011:**
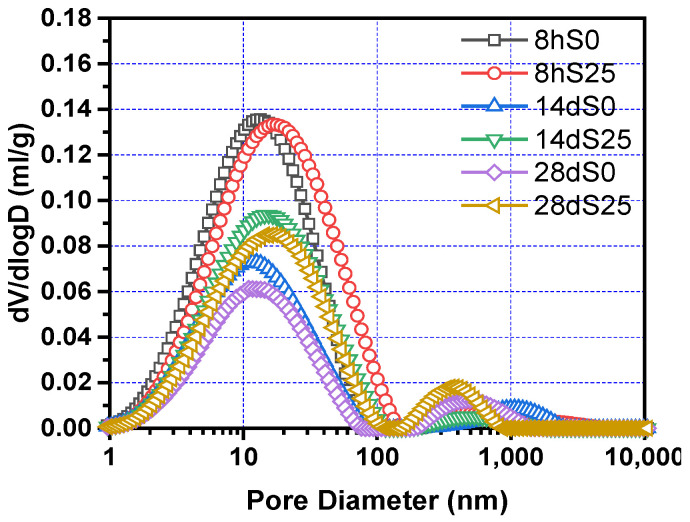
Pore size distribution of S0 and S25 at different times.

**Figure 12 materials-14-01647-f012:**
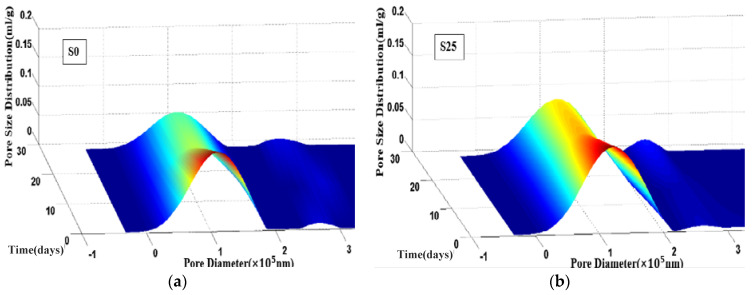
Development of pore size distribution of of S0 and S25 with age.

**Figure 13 materials-14-01647-f013:**
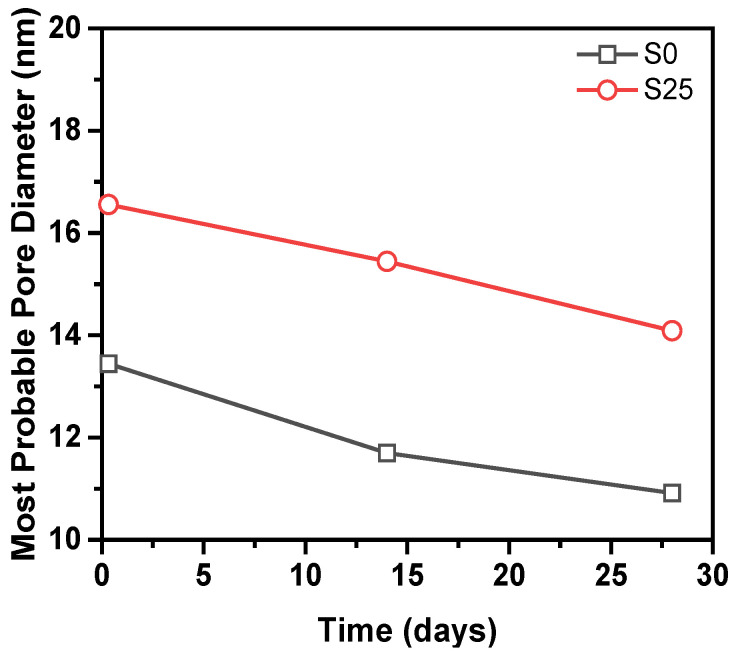
The most probable pore diameter of S0 and S25.

**Figure 14 materials-14-01647-f014:**
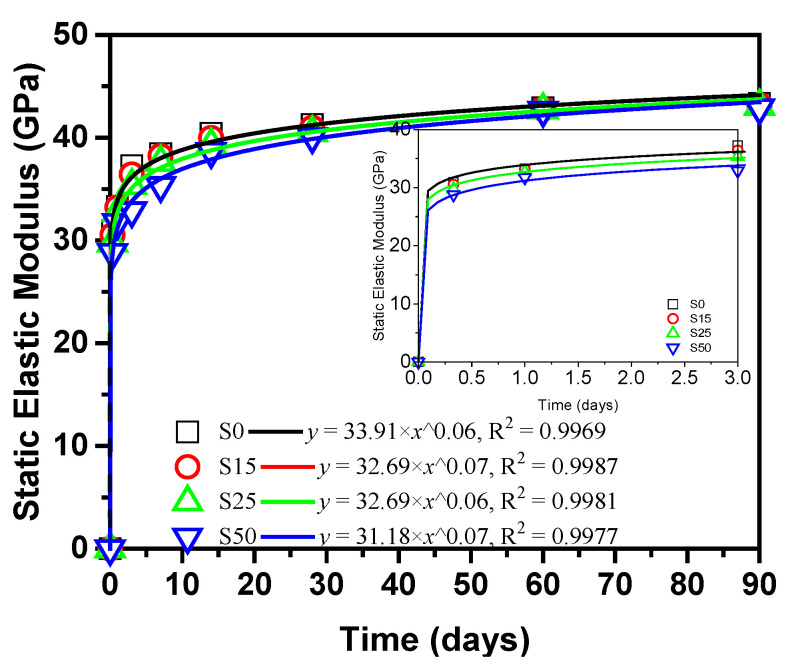
Elastic modulus of steam-cured concrete.

**Figure 15 materials-14-01647-f015:**
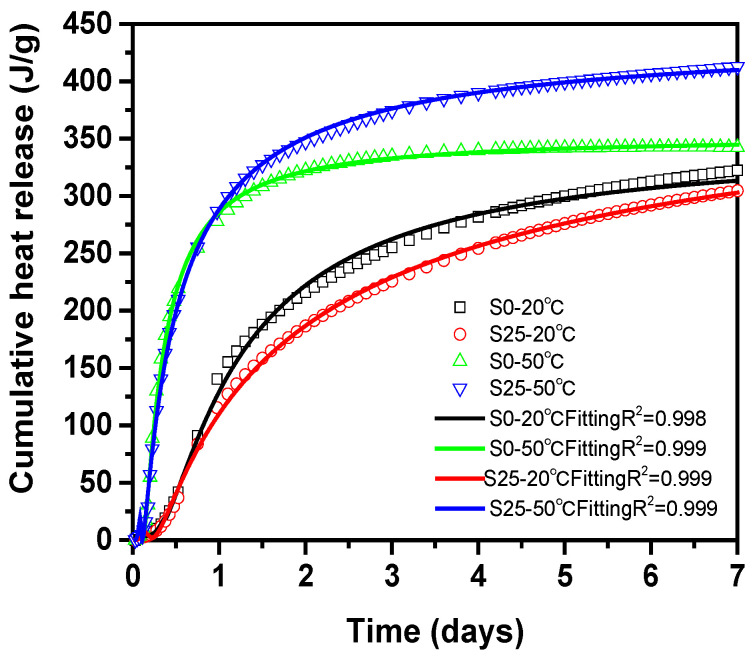
Cumulative heat release fitting.

**Figure 16 materials-14-01647-f016:**
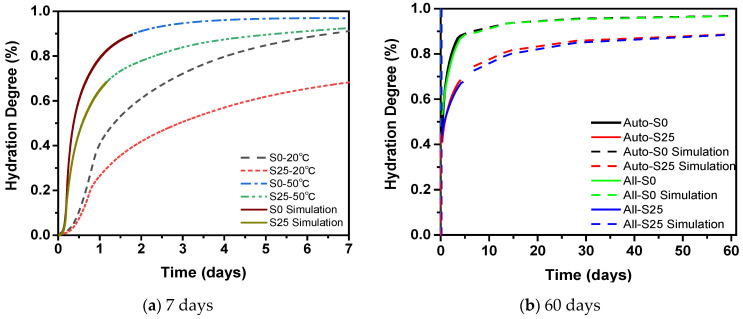
The hydration degree of steam-cured cement paste.

**Figure 17 materials-14-01647-f017:**
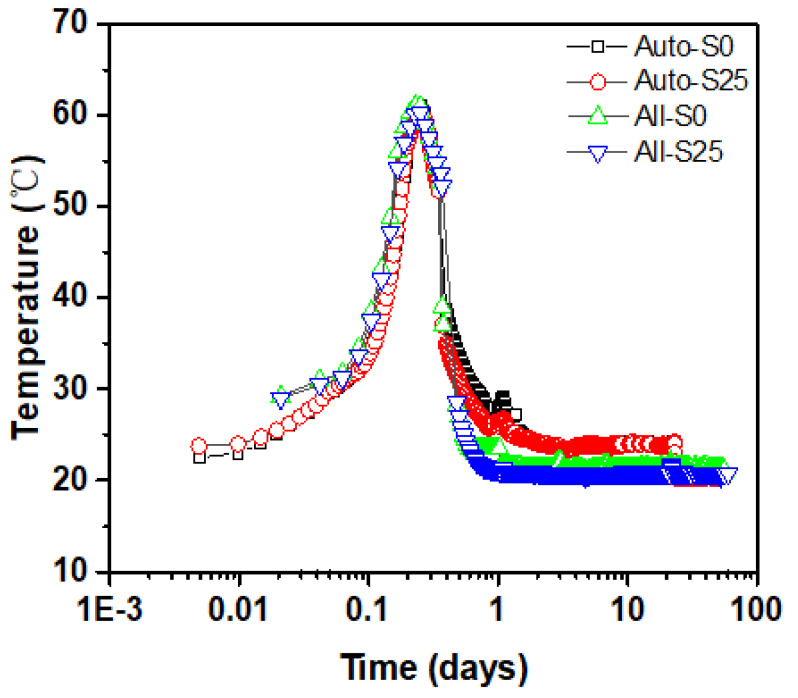
Temperature change inside steam-cured concrete.

**Figure 18 materials-14-01647-f018:**
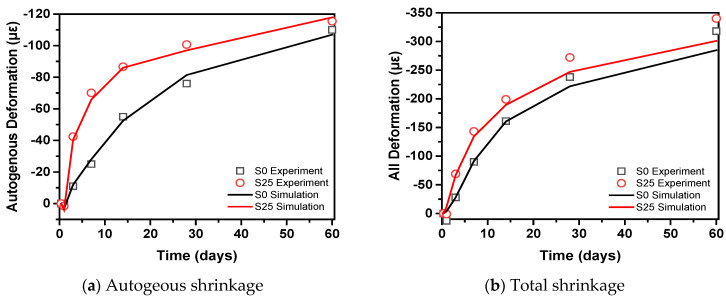
Comparison of experimental and calculated results of shrinkage of concrete.

**Table 1 materials-14-01647-t001:** Chemical composition of cementitious materials (%).

Raw Materials	SiO_2_	Al_2_O_3_	Fe_2_O_3_	CaO	MgO	SO_3_	TiO_2_	Na_2_O	K_2_O
Cement	19.95	4.83	2.93	65.71	2.95	2.28	0.34	0.21	0.8
Slag	30.15	15.53	0.43	41.94	7.89	2.22	0.82	0.56	0.46

**Table 2 materials-14-01647-t002:** Mix proportion of concrete (kg·m^−3^).

No.	Cement	Slag	Sand	Aggregate	Water	Superplasticizer
S0	460	0	780	1021	145	5.06
S15	391	69	780	1021	145	5.06
S25	345	115	780	1021	145	5.06
S50	230	230	780	1021	145	5.06

Note: S0——100%cement; S15——15%slag; S25——25% slag; S50——50% slag.

**Table 3 materials-14-01647-t003:** Theoretical relative humidity of saturated salt solution (%).

T (°C)	NaBr	NaCl	KCl	BaCl_2_	Distilled Water
20	59.1	75.5	85.1	90.0	100

**Table 4 materials-14-01647-t004:** Limit heat release (J/g).

Parameters	S0–20 °C	S0–50 °C	S25–20 °C	S25–50 °C
a_1_	350.250	418.202	496.181	498.779
a_2_	3.545	−64.782	−49.779	−49.857
a_1_ + a_2_	353.795	353.420	446.622	445.922

## Data Availability

Data are contained within the article.

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
