# Peer review of "Volume Deformation of Steam-Cured Concrete with Slag during and after Steam Curing"

_materials, 2021, doi:10.3390/ma14071647_

Round 1

Reviewer 1 Report

This paper studies the effects of the steam curing process and the content of slag on shrinkage deformation, hydration degree and elastic modulus of concrete. A very thorough set of experiments and analytical modelling is conducted. Some useful conclusions regarding shrinkage deformation and the use of slag are provided.
1. line 91 – should be workability
2. Many of the sub-headings have mixed use of italics – please be consistent and conform to journal standard
3. line 139 – “basically the same” is not scientific language
4. line 146 – should be semi-sealed
5. line 162 – how accurate is the assumption the pores are homogenous and cylindrical?
6. line 164 – equation 1 is shown twice?
7. lines 176, 178 – “is got” is not scientific language
8. Fig 5 – the references in the plot should be corrected to [X] format
9. line 204 – please define the “induction” period
10. line 237 – please define “a stable hole”
11. Fig 7 – the legend is too small to read
12. line 350 – please provide a reference for the “capillary pressure theory”
13. line 365 – what is meant by “move to the right”?
14. line 394 – please define Aft
15. line 452 – narrows – do mean over time?
16. it would be useful to add a sentence regarding the value of replacing cement with slag, based on your experimental results
17. There are some general English language errors that need to be carefully revised
18. Fig 14 - the inset is too small to read

Reviewer 2 Report

The topic is interesting and falls within the scope of the journal. Writting need a revision because sometimes the text is confusing and there are long-paragraphs that makes difficult the understanding.  

The applicability of the results and their influence in construction and civil engineering should be exposed. 

P2 L50-55 these statements need qualification, because the kind of described shrinkage just happens during the concrete setting and curing process at early ages 

P2 L60-62 When authors say "The cracks will reduce the durability of concrete and reduce the safety, reliability and service life of buildings. Especially buildings with large surface area and long-term exposure to the external environment" they must explain most clearly what are they talking about. What kind of cracking are they considering? Hydraulic process, thermal expansion / contraction? 

They should delimit which concrete ages are being considered. 

P2 L71-76 I agree, but the age is a very relevant factor that they must consider in the introduction 

P2 L81-82 "The obtained results can help understand and predict the shrinkage deformation of steam-cured concrete" During the curing process? Authors must clarify this question. 

2.2 Experimental tests. Authors may indicate the test sequence from the cast, the moulding, curing / steam curing process,... because  when they develope de methodology it result very confused 

2.1 Mixture proportions of concrete. The composition and information on the quality of the slag used should be provided 

P5 L143 demolished? 

2.2.4  It must be "Porometry"? 

P6 L162-163 “Assuming that the pores are homogeneous and cylindrical, the pore diameter d in the cement paste can be calculated according to formula (1)” A reference about this question must be provided. 

There is a point that authors should clarify. The porosimetry of concrete is not a very homogeneous value due to the characteristics of the material itself and the differences in the aggregate fractions. Considering that the cement paste behaviour will change when mixed with the agrgegate, what is the reason why the cement paste  is analysed before this process?

P7 195-197 The factors in the equations are not well expressed. 

Figure 5 is not referred in the text 

P8 Figure 5 must be Figure 6 

P8 L236 Figure number is wrong again 

Figure 7 is confused it should be divided in two or three figures 

Fig. 10 L356 Figure 10 

When talking about the pore size of concrete, it is difficult to obtain a homogeneous and representative sample, due to the heterogeneity of the material itself. 

What kind of sample has been used for these measurements? The high homogeneity obtained is really strange. 

According to my experience in porometry testing of mortar and concrete samples, these results are truly surprising. It is difficult to obtain such homogeneity and reproducibility in the curves. Authors must explain it widely. 

Equation 4 factors must be referred 

After the analysis carried out, it seems that the incorporation of slag does not favour the behaviour of the concrete, causing greater deformations due to shrinkage. Is this the case? 

References should be revised because they are not in accordance with the journal' standards. 

Reviewer 3 Report

The topic of research is up to date.
 The research area covered by the manuscript can be of potential interest for the readers of this Journal of Materials. 

The solved research area is Volume Deformation of Steam-Cured Concrete.

The experimental program is interesting, but the overall informational value of the manuscript needs to be improved.

Limitations of the experimental program are narrowly profiled tests. For a comprehensive description of the behaviour of concrete, it is necessary to determine more mechanical parameters, which also have a stochastic character.

Chapter 2 (2 Experimental program) - Move to page 3. 
Figure 4. - cannot read the descriptions in the image - that need to be edited.

Chapter 3.3  - Move to page 12. 

Figure 12. - cannot read the descriptions in the image - that need to be edited.

Figure 18. - cannot read the descriptions in the image - that need to be edited.

Please add a photo from the measurement/test samples.

Extensive research is underway in the solved problem of presented in manuscript and mechanical properties concrete. The part of the Introduction must be reworked and better stated the motivation and added value of the research carried out.

Possible sources include:
Sucharda, O; Mateckova, P. Bilek, V. Non-Linear Analysis of an RC Beam Without Shear Reinforcement with a Sensitivity Study of the Material Properties of Concrete. Slovak Journal of Civil Engineering 2020, 28 (1),  33-43. 
 Shah, M.I.; Amin, M.N.; Khan, K.; Niazi, M.S.K.; Aslam, F.; Alyousef, R.; Javed, M.F.; Mosavi, A. Performance Evaluation of Soft Computing for Modeling the Strength Properties of Waste Substitute Green Concrete. Sustainability 2021, 13, 2867.

There is a lack of criticism (recommendation) throughout the analysis of the information in the manuscript.

The manuscript is well organized and structured.

In the reviewer’s opinion, the manuscript has deficiencies that prevent its publication. 

The document must undergo revision.

Round 2

Reviewer 2 Report

This reviewer accepts the responses provided by the authors and consider that they have improved the first version

Reviewer 3 Report

Thank you for the adjustments made.
The changes made the improvement of the manuscript.

The research area and results are from the context of the manuscript can better understand.

The manuscript contains all the main information.

The manuscript can be published in the journal.